# p21^WAF1/Cip1^ Regulation by hYSK1 Activates SP-1 Transcription Factor and Increases MMP-2 Expression under Hypoxic Conditions

**DOI:** 10.3390/ijms20020310

**Published:** 2019-01-14

**Authors:** Mee-Hyun Lee, Joydeb Kumar Kundu, Bu Young Choi

**Affiliations:** 1China-US (Henan) Hormel Cancer Institute, No.127, Dongming Road, Jinshui District, Zhengzhou 450008, China; mhyun_lee@hanmail.net; 2Li Ka Shing Applied Virology Institute, University of Alberta, Edmonton, AB T6G 2R3, Canada; joydeb@ualberta.ca; 3Department of Pharmaceutical Science & Engineering, Seowon University, Cheongju 28674, Korea

**Keywords:** hYSK1, p21^WAF1/Cip1^, MMP-2, tumor migration, hypoxia

## Abstract

The hYSK1, a serine/threonine kinase (STK)-25, has been implicated in a variety of cellular functions including cell migration and polarity. We have recently reported that hYSK1 down-regulated the expression and functions of p16^INK4a^, a cell cycle regulatory protein, thereby enhancing migration and growth of cancer cells under hypoxic conditions. In this study, we further investigated the mechanisms underlying downregulation of p16^INK4a^ and anti-migratory function of hYSK1. Our study revealed that p21^WAF1/Cip1^ is a novel binding partner of hYSK1. Moreover, the interaction between hYSK1 and p21^WAF1/Cip1^ led to the inhibition of SP-1 transcriptional activity, as revealed by a significant down-regulation of SP-1-mediated transactivation of p16^INK4a^ promoter, and accelerated MMP-2 expression. Conversely, the knock-down of hYSK1 enhanced the p16^INK4a^ promoter activity and protein expression, and diminished MMP-2 transcription and protein levels in hypoxic conditions as compared to control. Taken together, hYSK1 blocks the p21^WAF1/Cip1^ functions by direct interaction and inhibits the p16^INK4a^ expression and induces MMP-2 expression by its regulations of SP-1 transcriptional activity under the hypoxia conditions.

## 1. Introduction

Despite relentless efforts in developing anticancer therapeutics, cancer still remains as the leading cause of death worldwide [1]. Although a wide variety of chemotherapy, radiotherapy and recently introduced immunotherapies are expected to cure cancer, there have been numerous reports of failure of these therapeutic modalities. One of the major reasons behind failure of anticancer therapies is the heterogeneous nature of the disease. Heterogeneity involves alterations, such as mutations, overexpression, loss or under-expression of various genes, especially those related to cell cycle regulation. For example, the impact of alterations in cell cycle checkpoint proteins, p16^INK4a^, p21^WAF1/Cip1^ and retinoblastoma (Rb) in tumor development and progression have been well investigated [2,3]. The p21^WAF1/Cip1^ and p16^INK4a^ act as tumor suppressors and arrest cell cycle progression by interfering with the interaction between cyclin-dependent kinases (Cdk) and cyclins [4]. Persistent intra-tumoral hypoxia is a characteristic feature of many solid tumors. The expression of both p21^WAF1/Cip1^ and p16^INK4a^ is increased in the early time point (~6 h) and then decreased in the later time point in various cancer cells under hypoxia condition [5]. The loss of p16^INK4a^ in primary tumors is attributed to various genetic and epigenetic alterations including mutation of the gene (CDKN2) encoding p16^INK4a^ protein or promoter hypermethylation [3]. Intra-tumoral hypoxia drives tumor cells to invade and migrate through stromal matrix and metastasize to distant locations, resulting in malignant carcinomas. The invasion and migration of tumor cells require alterations in cell polarization, activation of a group of matrix degrading enzymes, such as matrix metalloproteinases (MMP)-2 and -9. The transcriptional activation of MMP-2 is partly mediated by the binding of transcription factor specificity protein-1 (SP1) at the promoter site of MMP-2 and the activation of SP-1 is mediated via phosphorylation of SP-1 by cyclin-Cdks. The p16^INK4a^ as an inhibitor of Cdks have been shown to diminish SP-1 binding at MMP-2 promoter and reduced migration of cancer cells via downregulation of MMP-2 [6]. We have recently reported that the expression of p16^INK4a^ is negatively regulated by hYSK1, also known as a serine-threonine protein kinase (STK)-25, which plays a critical role in cell polarity, invasion, and migration. The hYSK1-mediated loss or downregulation of p16^INK4a^ has been shown to result in an increase in the expression of MMP-2 through transcriptional activation of SP-1, and in the enhancement of the migration and invasion of tumor cells including those of the melanoma and fibrosarcoma [7]. 

The p21^WAF1/Cip1^ is another Cdk inhibitor that suppresses tumor growth by inhibiting proliferation, inducing apoptosis, and blocking DNA synthesis in many cancers. Although p21^WAF1/Cip1^ being a p53-regulated gene is thought to play a tumor suppressive role, several studies have reported a pro-tumorigenic role of p21^WAF1/Cip1^ [8]. Zhang et al. [9] recently reported that the level of p21^WAF1/Cip1^is reduced in breast cancer tissues as well as in cultured cell lines, and the patients having high p21^WAF1/Cip1^ level showed longer survival than the patients having a low p21^WAF1/Cip1^ level. The micro-RNA (miR-3619)-mediated activation of p21^WAF1/Cip1^ induced growth arrest and inhibited metastasis of breast cancer cells [9]. Others have also shown that restoration of p21^WAF1/Cip1^ inhibits migration and invasion of various cancer cells [10,11]. Since hYSK1 enhanced tumor cells proliferation and migration by down-regulation of p16^INK4a^ [7], we have been interested to examine if p21^WAF1/Cip1^ also plays a role in hYSK1-induced tumor cells migration under hypoxia. The hYSK1 is an oxidative stress response protein. According to the Cancer Genome Atlas (TCGA) dataset, mRNA level of hYSK1 is high in cancers, such as glioma and endometrial cancer. In liver and colorectal cancers, a high protein expression of hYSK1 is associated with a decrease in five years survival rate as compared to those having a low expression of hYSK1 [12]. We have recently reported that hYSK1 interacted with 20–40 a.a. and 140–200 a.a. of p16^INK4a^, thereby decreasing the translocation of p16^INK4a^ from cytosol to nucleus in hypoxic condition [7]. The reduced p16^INK4a^ nuclear localization by overexpression of hYSK1 leads to the activation of SP-1 transcriptional activity, resulting in increased MMP-2 expression and enhanced cancer cell proliferation and migration. Since p21^WAF1/Cip1^ inhibits cyclin-Cdks, we were interested to see if hYSK1 also targets p21^WAF1/Cip1^ to excel tumor cell migration via activation of MMP-2. Here we report that hYSK1 directly interacted with p21^WAF1/Cip1^, thereby resulting in SP-1-mediated suppression of p16^INK4a^ promoter activity and increased MMP-2 expression under hypoxic condition leading to increased proliferation and migration of cancer cells.

## 2. Results

### 2.1. hYSK1 Interacts with p21^WAF1/Cip1^

We attempted to assess the interaction between hYSK1 and p21^WAF1/Cip1^ by GST pull-down analysis. The ^35^S-labeled p21^WAF1/Cip1^ proteins were incubated with either GST or GST-hYSK1 fusion proteins bound to Sepharose beads and precipitated in a pull-down assay. Results indicated that GST-hYSK1 was bound to p21^WAF1/Cip1^, whereas GST alone did not bind (Figure 1).

### 2.2. Interaction between hYSK1 and p21^WAF1/Cip1^ Regulates SP-1 Transcriptional Activities

We first examined the role of p21^WAF1/Cip1^ in p16^INK4a^ expression by assessing the SP-1 transcription factor activity through the p16^INK4a^ promoter luciferase assay (Figure 2). As shown in Figure 2, with the increasing amount of p21^WAF1/Cip1^ SP-1 transcriptional activity, as revealed by SP-1-mediated p16^INK4a^ luciferase activity, was significantly induced (Figure 2). To identify the role of hYSK1 on p16^INK4a^ expression by regulation of p21^WAF1/Cip1^ and SP-1, we transfected hYSK1 under the *p16^INK4a^* promoter luciferase gene, *SP-1* and *p21^WAF1/Cip1^* together (Figure 3A). The results showed that p16^INK4a^ gene transcription was induced by transfection of SP-1 and it is enhanced by transfection of p21^WAF1/Cip1^, however p16^INK4a^ gene transcription was decreased by transfection of hYSK1 in an amount-dependent manner (Figure 3A). This phenomenon was verified by immunoprecipitation analysis (Figure 3B). After immunoprecipitation with anti-hYSK1, we detected p21^WAF1/Cip1^ (Myc tag) but not SP-1 (Figure 3B). Conversely, hYSK1 expression was not detected but p21^WAF1/Cip1^ expression was observed after immunoprecipitation with anti-SP-1 (Figure 3B). It means that the interaction between hYSK1 and p21^WAF1/Cip1^ causes to block the SP-1 transcriptional activity for p16^INK4a^ transcription.

### 2.3. Regulation of SP-1 Transcriptional Activities by Interaction of hYSK1 and p21^WAF1/Cip1^ Increases MMP-2 Transcription and Accelerates Cell Migration under Hypoxia Conditions

In the hypoxia condition (1% O_2_), we found that the SP-1 transcriptional activity was decreased against p16^INK4a^ promoter in a time-dependent manner (Figure 4A). For knock down of hYSK1, SK-MEL-28 cells were transfected with *siRNA-hYSK1* and the cells were incubated in 5% CO_2_ incubator for 12 h then moved to 1% O_2_ hypoxic chamber and incubated for 0, 6 or 24 h (Figure 4B). The p16^INK4a^ transcription was increased at 24 h under hypoxia as compared to transfection with *siRNA-control* or *pGL2 control* (Figure 4B). However, MMP2 transcription was decreased at the same time and conditions (Figure 4B). We also examined the expression of above genes in hypoxic conditions by time dependent manner (Figure 5A). The results revealed that the expression of hYSK1, MMP-2 and HIF-1α was increased and that of p21^WAF1/Cip1^ was decreased. The p16^INK4a^ expression was increased at 6 h and decreased at 24 h. Interestingly, we precipitated the cell lysate by SP-1 antibody and phosphorylated SP-1 expression was increased but intact form of SP-1 was decreased in hypoxic condition by the time course (Figure 5A). When hYSK1 was deficient by transfection of siRNA-hYSK1, the increased hYSK1 and MMP-2 expression by siRNA-control was disrupted in hypoxia condition (Figure 5B). However, p16^INK4a^ expression was increased by knock-down of hYSK1 compared to control. Although p21^WAF1/Cip1^ expression was not much different, phosphorylated Rb expression was markedly decreased in hypoxia condition (Figure 5B). The promoter activity of *p16^INK4a^* is attenuated by hYSK1 through the interaction of hYSK1 with p21^WAF1/Cip1^ and subsequent blockade of p21^WAF1/Cip1^- and SP-1-mediated transactivation of p16^INK4a^. To address the functional significance of hYSK1 and p21^WAF1/Cip1^, we have conducted real-time cell migration analysis (Figure 5C,D). After knockdown of hYSK1, the cell migration was decreased in a time-dependent manner as compared to control cells under hypoxic condition (Figure 5C). Also when we silenced p21^WAF1/Cip1^ in SK-MEL-28 cells, the cell migration was significantly increased under hypoxia condition compared to normoxia, and it was abolished by knockdown of hYSK1 (Figure 5D).

## 3. Discussion

The perturbation of cell cycle checkpoint proteins, especially cyclin-Cdk inhibitor’s function is common in many cancers. Among the cyclin-Cdk inhibitors, the p16^INK4a^ and p21^WAF1/Cip1^ have been extensively studied for their role in neoplastic transformation of cells. The p16^INK4a^ alteration has caused by genetic and epigenetic regulation such as promoter hyper-methylations and histone modifications [13,14]. The p16^INK4a^ is involved in arresting cell cycle at G0 and early G1 phase by disrupting cyclin-Cdk4/6 interaction. The inactivation of Cdk4/6 activity results in reduced phosphorylation of pRb and compromised interaction between phospho-pRb and E2F transcription factor, thereby reducing the transcription of genes encoding proteins involved in cell proliferation [15]. Besides blocking cyclin-Cdk4/6-mediated pRb phosphorylation, p16^INK4a^ also causes cell cycle arrest by binding with transcription factor TFIIH, resulting in reduced phosphorylation of RNA polymerase II [16]. It is also reported that p16^INK4a^ interacted with c-Jun N-terminal kinases (JNKs) and inhibits the UV-induced cell transformation by activation of c-Jun transcription factor [17]. We have previously reported that p16^INK4a^ function is lost by its interaction with eukaryotic elongation factor 1A2 and hYSK1, thereby increasing proliferation and migration of various cancer cells [7,18]. The p21^WAF1/Cip1^ is another cell cycle regulatory protein that also inhibits cyclin dependent kinases by working as a downstream signaling molecule to p53. Although initially identified as a kinase inhibitor to induce apoptosis, there is paradoxical findings that p21^WAF1/Cip1^ allows cell cycle exit and promotes cell proliferation [8]. It has been reported that p21^WAF1/Cip1^ regulates the expression of p16^INK4a^ by binding of SP-1 transcription factor on GC region (−449~−459) of p16^INK4a^ promoter [19]. 

The hYSK1, also known as STK25, is a member of the germinal center kinase III (GCK III) subfamily of the sterile 20 (STE20) kinase superfamily [20]. Although much is known about the role of this protein in targeting Golgi apparatus, in the regulation of cellular polarization, program cell death and cell migration under stress conditions [21,22], role of this protein in cancer has not fully elucidated. We have recently reported that hYSK1 promotes migration of melanoma and fibrosarcoma cells by repressing p16^INK4a^ nuclear translocation via direct interaction, thereby resulting in increased SP-1-mediated MMP2 transcription [7]. In the present study, we sought to interrogate if hYSK1 follows additional mechanisms in promoting tumor cell proliferation and migration. We have identified p21^WAF1/Cip1^ as a novel binding partner of hYSK1 (Figure 1). Moreover, by increasing the amount of p21^WAF1/Cip1^, we noticed the enhancement of SP-1-mediated p16^INK4a^ transactivation (Figure 2). Furthermore, the p16^INK4a^ expression was inhibited by increasing amount of hYSK1 because hYSK1 has interacted with p21^WAF1/Cip1^ but not SP-1, and reduced the binding of SP-1 on the p16^INK4a^ promoter (Figure 3). 

Hypoxia condition is the main signature of cancer mass and promotes epithelial-mesenchymal transition, migration and invasion of cancer cells [23,24]. Among the many different signaling pathways involved in hypoxia-triggered tumor cell migration, hYSK1 has recently been identified as a potential regulator, which diminishes p16^INK4a^ function. Our study revealed that the binding activity of SP-1 on p16^INK4a^ promoter was decreased in hypoxia condition by time course (Figure 4). We also showed that the hYSK1 inhibited p16^INK4a^ expression but induced MMP-2 expression and cell migration in hypoxia condition (Figure 5). Taken together, we identified a potential regulatory mechanism of p16^INK4a^/MMP-2 expression in cancer by loss of function of p21^WAF1/Cip1^ through the interaction with hYSK1 in hypoxia (Figure 6). Conversely, Wu et al. [25] have demonstrated that STK25 (a.k.a hYSK1) suppressed the proliferation of human colorectal cancer cells and silencing STK25 enhanced xenograft tumor growth in mice. Thus, additional studies are warranted to define the exact role of hYSK1 in cancer. However, based on the result of present study, the antagonist development of hYSK1 would be considered as a new therapeutic avenue for developing anticancer drugs.

## 4. Materials and Methods

### 4.1. Cell Culture, Plasmids, and Transfection

SK-MEL-28 (human melanoma), HT-1080 (fibrosarcoma) and COS-7 (African green monkey kidney) cells were purchased from the American Type Culture Collection (ATCC, Manassas, VA, USA). SK-MEL-28 cells and HT-1080 were cultured in MEM containing penicillin (100 units/mL), streptomycin (100 μg/mL), sodium pyruvate (1 mM), and 10% fetal bovine serum (FBS) (Invitrogen, Gibco, MA, USA). COS-7 cells were grown in DMEM containing penicillin (100 units/mL), streptomycin (100 µg/mL), and 10% FBS. Cells were maintained at 37 °C in a humidified atmosphere of 95% air/5% CO_2_.

*pcDNA3.1-v5-hYSK1* was generated by PCR using the human cDNA clone-*hYSK1* (Origene Technologies, Rockville, MD, USA) as a template. The PCR product was purified, digested with *EcoRI/XhoI*, and cloned into the *EcoRI/XhoI* sites of *pcDNA3.1-v5-HisA* (Invitrogen). The *GST-hYSK1* was inserted in-frame into the *BamHI/XhoI* site of the *pGEX-5X-1* vector (Amersham Biosciences, Buckinghamshire, UK). The *HA-SP-1* and *Myc-p21^WAF1/Cip1^* were cloned into the *EcoRI/XhoI* site of the *pCMV-HA* and *pCMV-Myc* vectors (Clontech Laboratories, Mountain View, CA, USA). The *pGL2-p16^INK4a^-luc* vector was a gift from Dr. Gordon Peters (Cancer Research UK, London, UK) and the *pGL2-MMP-2-luc* vector was a gift from Dr. Etty N. Benveniste (The University of Alabama at Birmingham, Birmingham, AL, USA). Various expression vectors were amplified in *E. coli* XL1-blue or BL21 cells and plasmids were purified using a Qiagen midi kit (Qiagen, Hilden, Germany). The DNA sequences of all plasmids were confirmed by sequencing (Dye Terminator ABI Type Seq., Bionex, NJ, USA).

COS-7 cells were transfected with *pGL2-p16^INK4a^-luc* and, *pcDNA3.1- p21^WAF1/Cip1^*, *HA-SP-1* or *pcDNA3.1-v5-hisA-hYSK* plasmids using the jetPEI poly transfection reagent (Polyplus, Illkirch, France). For transient gene silencing, SK-MEL-28 cells were transfected using *si-control* and *si-hYSK1* (Dharmacon, Lafayette, CO, USA). For the reporter gene assay, COS-7 cells were seeded in 24-well plates and incubated for 24 h followed by transfection with, *pGL2-p16-luc*, or *pGL2-MMP-2-luc* reporter plasmids using the jetPEI poly transfection reagent based on the manufacturer’s instructions. The *pRL-TK* reporter plasmid was used as an internal control. Cells were harvested after 24 h and disrupted with 5× lysis buffer. Luciferase activity was determined after normalization to *pRL-TK* activity (Promega, Madison, WI, USA).

### 4.2. GST Pull-Down Assay

Full length of p21^WAF1/Cip1^ was translated in vitro with l-[^35^S] methionine using the TNT Quick Coupled Transcription/Translation System (Promega). Full-length YSK1 proteins were produced in *E. coli* BL21 as GST-fusion proteins and then purified on GST-Sepharose 4B beads. The ^35^S-Met-labeled p21^WAF1/Cip1^ proteins were incubated with either GST or GST-YSK1 fusion proteins bound to Sepharose beads and precipitated in a pull-down assay. The bound proteins were washed three times and boiled with 2.5× sample buffer for 3 min, centrifuged, and then the supernatant fraction was examined by 15% SDS-PAGE analysis. The binding was detected by autoradiography.

### 4.3. Transfection and Luciferase Activity

The *pcDNA3.1-p16^INK4a^* or *pcDNA3.1-v5-hisA-hYSK1* plasmid was transfected using the jetPEI poly transfection reagent (Polyplus, Illkirch, France) into HT-1080, COS-1, or COS-7 cells to generate p16^INK4a^- or p16^INK4a^-hYSK1-expressing cells. For transient gene silencing, *si-control* and *si-hYSK1* (Dharmacon) were transfected into HT-1080, SK-MEL-28, and A2058 cells. *shRNA-control* and *shRNA-hYSK1* were transfected into HT-1080 and SK-MEL-28 cells. For the reporter gene assay, COS-7 cells were seeded in 24-well plates and incubated for 24 h followed by transfection with *pcDNA3-p16^INK4a^*, *HA-SP-1*, *cyclin A*, *pGL2-p16-luc*, or *pGL2-MMP-2-luc* reporter plasmids using the jetPEI poly transfection reagent based on the manufacturer’s instructions. The *pRL-TK* reporter plasmid was used as an internal control. Cells were harvested after 24 h and disrupted with 5× lysis buffer. Luciferase activity was determined after normalization to *pRL-TK* activity (Promega).

### 4.4. Electrophoretic Mobility Shift Assay (EMSA)

EMSA for SP-1 DNA binding was performed using a DNA–protein binding detection kit, according to the manufacturer’s protocol (GIBCO BRL, Grand Island, NY, USA). Briefly, the SP-1 oligonucleotide probe 5′–GACCGAGTGCGCTCGGCGGCTGCGGAGAGGGGTAGAGCAGGCAGCGGGCGGCGGGGAGCAGC–3′ (SP-1 binding site in p16^INK4a^ promoter) [26] was labeled with [γ-^32^P]ATP by T4 polynucleotide kinase and purified on a Nick column (Amersham Pharmacia Biotech, Buckinghamshire, UK). The binding reaction was carried out in 25 mL of the mixture containing 5 mL of incubation buffer (10 mM Tris–HCl, pH 7.5, 100 mM NaCl, 1 mM DTT, 1 mM EDTA and 4% glycerol), 10 mg of nuclear extracts and 100,000 c.p.m. of [γ-^32^P] ATP-end labeled oligonucleotide. After 50 min incubation at room temperature, 2 mL of 0.1% bromophenol blue was added, and samples were separated through 6% non-denaturing polyacrylamide gel at 150 V in a cold room for 2 h. Finally, the gel was dried and exposed to an X-ray film.

### 4.5. Immunoprecipitation

Transfected COS-7 cells or SK-MEL-28 cells were harvested in NET-NL lysis buffer containing 50 mM Tris (pH 7.5), 5 mM EDTA, 150 mM NaCl, 1 mM DTT, 0.01% NP-40, 0.2 mM PMSF, and a mixture of protease inhibitors (Roche Diagnostics, Basel, Switzerland). Cell lysates (200 μg) were clarified by centrifugation before overnight incubation at 4 °C with hYSK1 and SP-1 (Santa Cruz Biotechnology, Dallas, TX, USA) antibodies in NET-NR buffer (50 mM Tris (pH 7.5), 5 mM EDTA, 150 mM NaCl, 1 mM DTT, 0.01% NP-40, 2 μg/mL BSA, 0.2 mM PMSF), and a mixture of protease inhibitors (Roche Diagnostics, Basel, Switzerland). An aliquot of 50 μL pre-washed protein A/G-agarose beads (Roche Diagnostics; 50% slurry) was then added to the mixture and incubated for 2 h at 4 °C. Immunoprecipitates were recovered by centrifugation, washed three times in NET-NW buffer (50 mM Tris (pH 7.5), 5 mM EDTA, 150 mM NaCl, 1 mM DTT, 0.01% NP-40, and 0.2 mM PMSF) and resolved by sodium dodecylsulfate (SDS)-polyacrylamide gel electrophoresis (PAGE) and western blotting.

### 4.6. Western Blot Analysis

Cells were disrupted on ice for 30 min in cell lysis buffer which contains 20 mM Tris (pH 7.5), 150 mM NaCl, 1 mM Na_2_EDTA, 1 mM EGTA, 1% Triton X-100, 2.5 mM sodium pyrophosphate, 1 mM β-glycerophosphate, 1 mM sodium vanadate, 1 μg/mL leupeptin, and 1 mM phenylmethylsulfonyl fluoride. After centrifuged at 14,000 rpm for 20 min, the cell lysates were collected, and protein concentration was determined by using a protein assay reagent (Bio-Rad labs, Hercules, CA, USA). The total cellular protein extracts were separated by SDS-PAGE and transferred to polyvinylidene fluoride (PVDF) membranes in 20 mM Tris-HCl (pH 8.0), containing 150 mM glycine and 20% (*v*/*v*) methanol. Membranes were blocked with 5% nonfat dry milk in 1× Tris-buffered saline (TBS) containing 0.05% Tween 20 (TBS-T) and incubated with antibodies against p16^INK4a^, p21^WAF1/Cip1^, hYSK1, pRb, Rb, HIF-1a, pSP-1, SP-1, MMP-2, and β-actin. The blots were washed three times in 1× TBS-T buffer followed by incubation with the appropriate HRP-linked IgG. The specific proteins in the blots were visualized using an enhanced chemiluminescence (ECL) detection kit (GE Healthcare Bioscience, Pittsburgh, PA, USA).

### 4.7. Cell Migration Assay

Wound migration of cells was measured using Culture-Inserts (Ibidi GmbH, Martinsried, Germany). The Culture-Inserts were placed in 30 mm dish, and *siRNA-control* and *siRNA-hYSK1*-transfected SK-EML-28 cells were seeded at a density of 5 × 10^4^ cells in each well with the Culture-Inserts. After 24 h of incubation, the Culture-Inserts were removed, a cell-free gap of 500 μm was created and then transferred to a hypoxic (1% CO_2_) chamber. The movement of cells was detected by an inverted microscope (BX50, Olympus, Tokyo, Japan) by time course.

Cell movement was analyzed using AVI meta imaging software. For real-time cell migration, we used the xCELLigence RTCA DP system and fibronectin-coated CIM plates (Roche Diagnostics, Basel, Switzerland). Cells were seeded at 10,000 cells per well and the migratory behavior of each cell line was monitored for 30 h. The assay was performed based on the manufacturer’s instructions (Roche Diagnostics, Basel, Switzerland).

### 4.8. Statistical Analysis

Values were expressed as means ± S.E.M. from at least three independent experiments. Statistical significance was determined by Student’s *t*-test and a *p*-value less than 0.05 was considered statistically significant.

## Figures and Tables

**Figure 1 ijms-20-00310-f001:**
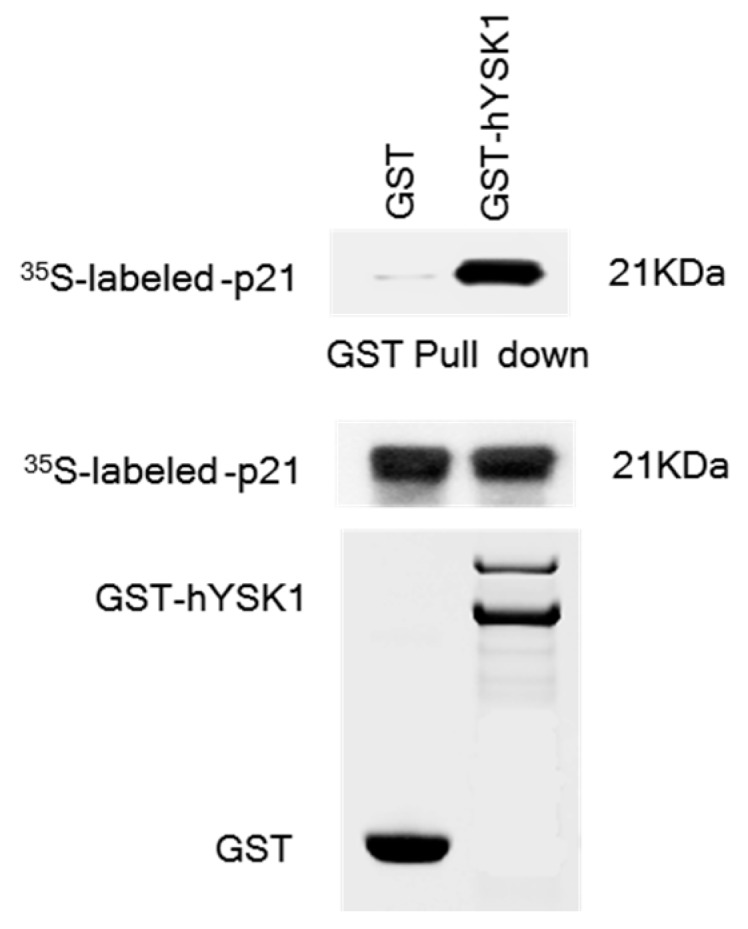
hYSK1 interacts with p21^WAF1/Cip1^. GST-hYSK1 interacts with in vitro transcribed and translated p21^WAF1/Cip1^. After pull-down with GST-hYSK1, western blots of in vitro translated and bound proteins are shown (upper panel). p21^WAF1/Cip1^ expression represents that even amount of p21^WAF1/Cip1^ protein incubated with GST-hYSK1 or GST only (middle panel). Coomassie blue-stained gels showed GST–hYSK1 and GST as a loading control (lower panel).

**Figure 2 ijms-20-00310-f002:**
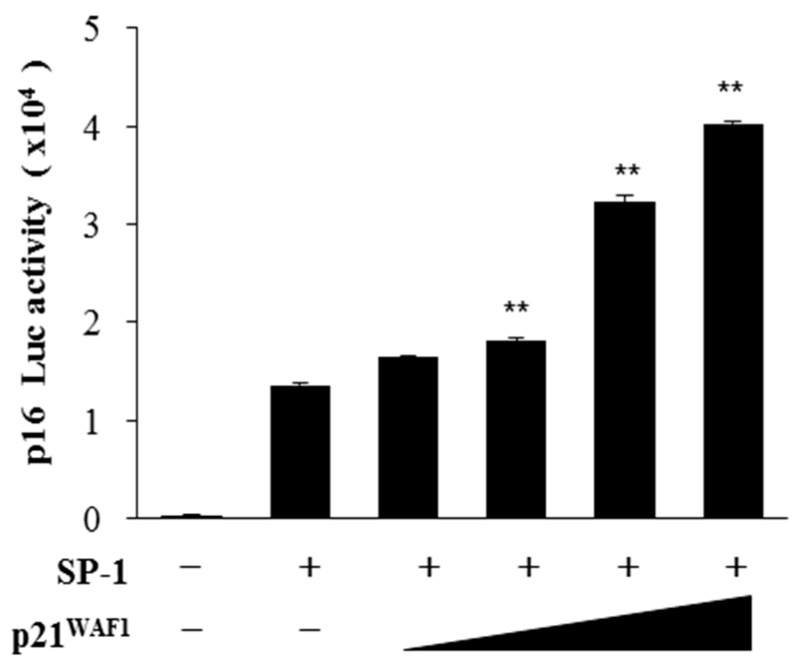
p21^WAF1/Cip1^ up-regulates the SP-1 transcriptional activity on p16^INK4a^ promoter. Transfection of SP-1 and increased amount of p21^WAF1/Cip1^ was accessed by jetPEI poly transfection reagent on COS-7 cells. Transcriptional activity of SP-1 was determined by p16^INK4a^ promoter luciferase activities. Relative p16 luciferase activity was determined (*n* = 6, data shown as mean ± SEM, ** *p* < 0.05 as determined by paired *t*-test).

**Figure 3 ijms-20-00310-f003:**
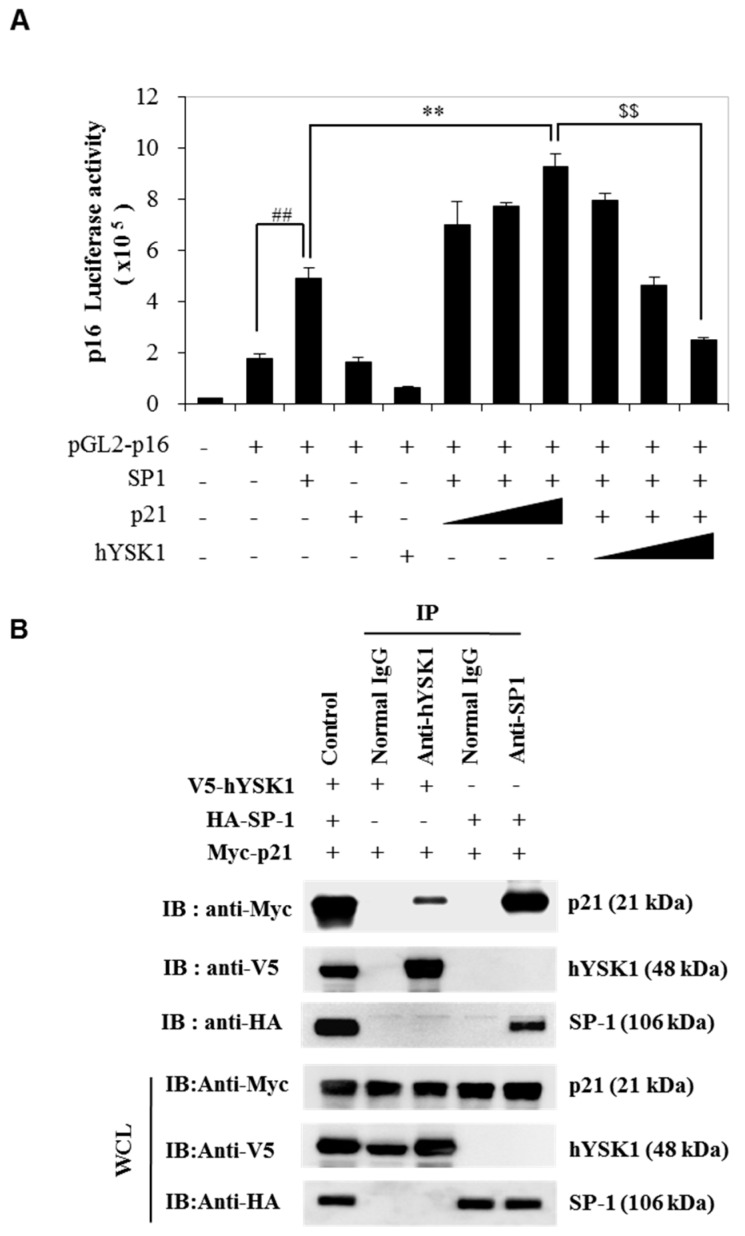
hYSK1 down-regulates SP-1 transcriptional activity of p16^INK4a^ by interaction with p21^WAF1/Cip1^. (**A**) The effect of hYSK1 on p16^INK4a^ expression was determined by luciferase activity with transfection of *pGL2-p16-luc*, *SP-1*, *p21* and *pcDNA3.1-V5-hYSK1* in COS-7 cells. p16 promoter activity was determined by luciferase assay (*n* = 6, data shown as mean ± SEM, ^##^, ** and ^$$^
*p* < 0.05 as determined by paired *t*-test). (**B**) Immunoprecipitation was performed using anti-hYSK1 or SP-1 in cells and performed western blotting with target antibodies, p21 (Myc), hYSK1 (V5) or SP-1 (HA).

**Figure 4 ijms-20-00310-f004:**
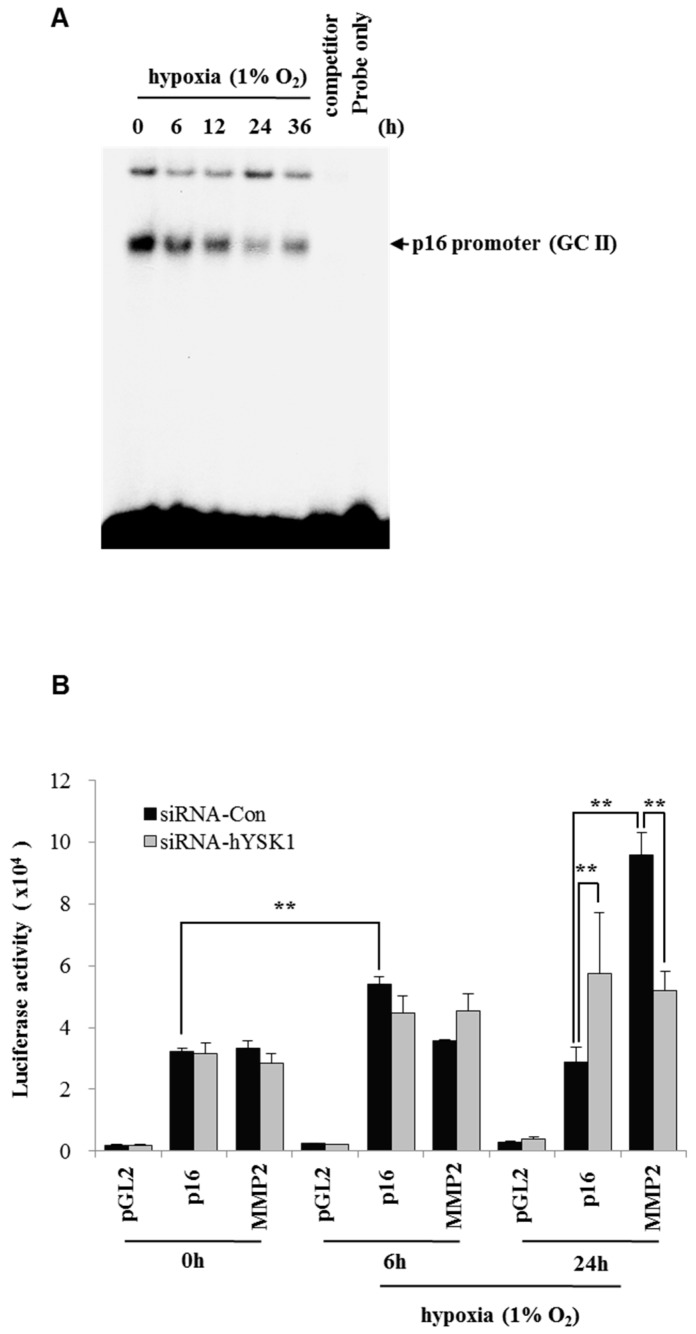
hYSK1 regulates the transcription of p16^INK4a^ and MMP-2 in hypoxia. (**A**) The effect of SP-1 binding on the p16^INK4a^ promoter under hypoxic conditions (1% O_2_). Nuclear fraction of cell lysate was separated by EMSA analysis. (**B**) The effect of hYSK1 on p16^INK4a^ and MMP-2 transcription. After transfection of *siRNA-control*, *siRNA-hYSK1*, *pGL2-p21-luc* and *pGL2-MMP-2-luc* in HT1080 cells, cells were incubated in hypoxia condtion for 24 h and measured the luciferase activities (*n* = 6, data shown as mean ± SEM, ** *p* < 0.05 as determined by paired *t*-test).

**Figure 5 ijms-20-00310-f005:**
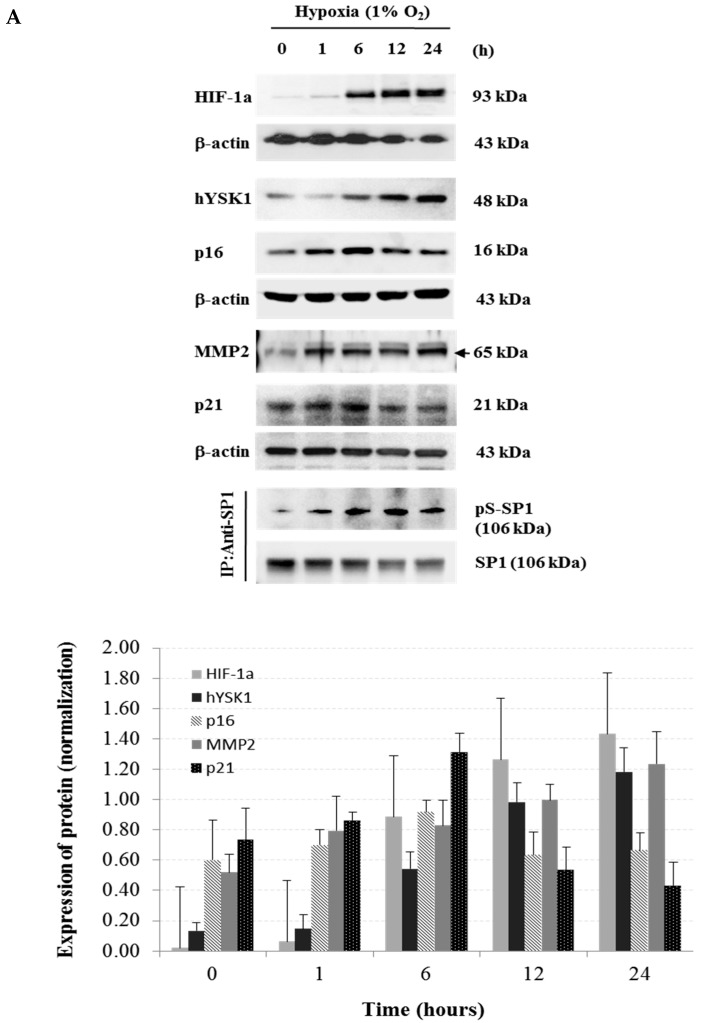
hYSK1 regulates the expression of p16^INK4a^ and MMP-2 in hypoxia. (**A**) The expression of HIF-1α, hYSK1, p16^INK4a^, p21^WAF1/Cip1^ and MMP-2 under hypoxic (1% O_2_) conditions by time course. (**B**) The effect of hYSK1 on p16^INK4a^ and MMP-2 expression. After transfection of *siRNA-control* and *sRNAi-hYSK1* in HT1080 cells, cells were incubated under hypoxia condition for 0, 6, 12 and 24 h, and the expression of indicated proteins was detected by western blotting. (**C**) Cell migration after knockdown of hYSK on SKMEL28 cells under hypoxia. The pictures represented the closed gap of SKMEL28 cell with *siRNA-control* or *siRNA-hYSK1* at 2 h, 20 h and 30 h. (**D**) Cell migration index after knockdown of *hYSK*, *p21^WAF1/Cip1^* or both of *hYSK* and *p21^WAF1/Cip1^* in SKMEL28 cells. Cells were seeded at 10,000 cells per well in fibronectincoated CIM 96 well plates and the migratory behavior of cell was monitored for 30 h under normoxia and hypoxia. (*n* = 6, data shown as mean ± SEM, ** *p* < 0.05 as determined by paired *t*-test).

**Figure 6 ijms-20-00310-f006:**
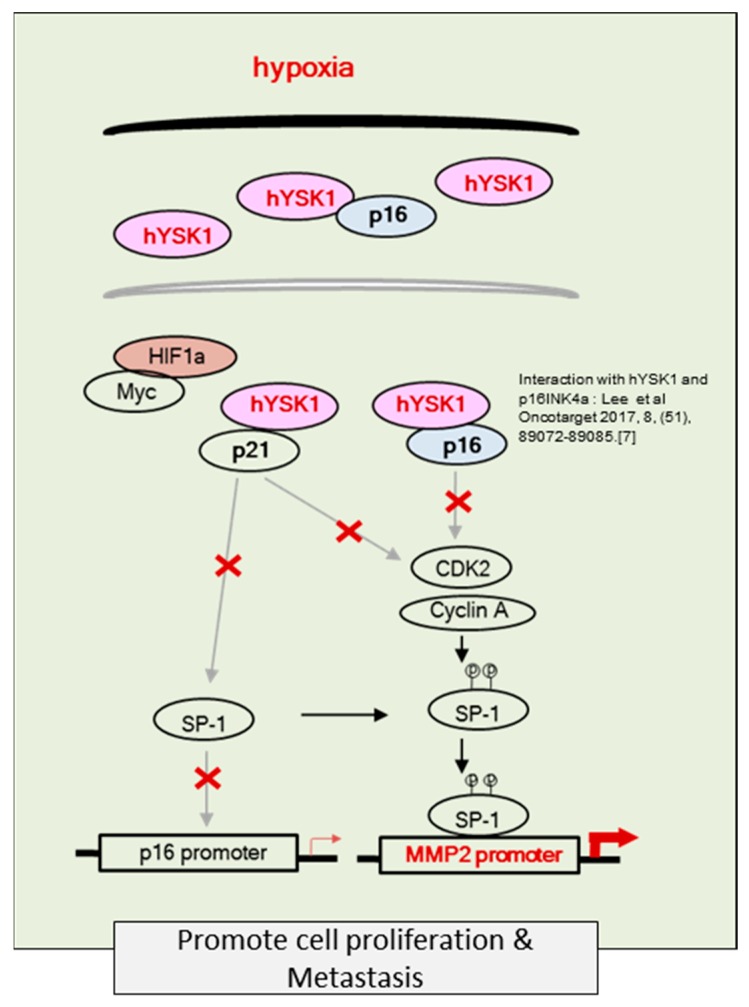
Schematic regulation of p16^INK4a^ and MMP-2 expression by hYSK1. A representative scheme of hYSK1 regulation of p16^INK4a^ or p21^WAF1/Cip1^ function which inhibits CDKs/cyclins and cell proliferation by directly interactions. Furthermore, interaction of hYSK against p21^WAF1/Cip1^ caused to decrease of p16^INK4a^ transcription by SP-1 transcription factor, thereby activating cell proliferation, migration and metastasis. Interaction results of hYSK1 and p16 were published earlier [7].

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
