# Peer review of "p21^WAF1/Cip1^ Regulation by hYSK1 Activates SP-1 Transcription Factor and Increases MMP-2 Expression under Hypoxic Conditions"

_ijms, 2019, doi:10.3390/ijms20020310_

Round 1
Reviewer 1 Report
Review for IJMS
Title of Manuscript: p21WAF1/Cip1 regulation by hYSK1 activates SP-1 transcription factor and increases MMP-2 expression under hypoxic conditions.
Recommendation: Major revision
This article describes the interaction of hYSK1 with p21 and blocks SP-1 transcription activity on the p16 promoter thereby accelerating MMP-2 under hypoxia conditions.
The manuscript has merit but major flaws in the presentation and experiments.
Major comments.
Comment 1: The authors major message is the interaction occurs under hypoxic conditions. At no point in the methods or results do the authors provide the protocol used. This must be clarified in the methodology.
Comment 2: Figure 5 shows western blots highlighting the major findings in the paper.
1) There is no indication of the size of any of the proteins.
2) There is only one b-actin control for each panel of Western blots. It is clear that the b-actin controls and the proteins above are not from the same Western blots.
3) How many times were the experiments done? These experiments should be done in triplicate as they are central to the results.
4) There are no statistics for any of the blots shown.
Comment 3: The manuscript is very poorly written with many grammatical errors, long vague, incorrect statements, and poor sentence construction throughout the text - too many to list. There are sloppy spelling mistakes which a simple spell check should pick up.
One example: starting line 29: According to a statistics, the incidence of about 1,735,350 new cancer cases and 609,640 cancer-related deaths have been recorded only in the United States, this year [2]. Does this mean that cancer is ‘only’ in the US??
Author Response
Point-by-Point Response to Reviewer’s Comments:
Manuscript ID ijms-405551
Title : p21WAF1/Cip1 regulation by hYSK1 activates SP-1 transcription factor and increases MMP-2 expression under hypoxic conditions
Reply to Reviewer #1:
This article describes the interaction of hYSK1 with p21 and blocks SP-1 transcription activity on the p16 promoter thereby accelerating MMP-2 under hypoxia conditions.
The manuscript has merit but major flaws in the presentation and experiments.
Major comments
Comment 1: The authors major message is the interaction occurs under hypoxic conditions. At no point in the methods or results do the authors provide the protocol used. This must be clarified in the methodology.
Response 1: We appreciate this critical suggestion. We have detailed the experimental protocols specifying the hypoxia conditions in revised methods and legend sections.
Comment 2: Figure 5 shows western blots highlighting the major findings in the paper.
Comment 2-1 There is no indication of the size of any of the proteins.
Response 2-1: We have indicated the size of proteins in Figures
Comment 2-2 There is only one b-actin control for each panel of Western blots. It is clear that the b-actin controls and the proteins above are not from the same Western blots.
Response 2-2: Thank you for pointing out this error. We revised the b-actin control we had got from respective immunoblot membrane. In some cases the b-actin band is same for two or more proteins of different size detected from the same membrane.
Comment 2-3 How many times were the experiments done? These experiments should be done in triplicate as they are central to the results.
Response 2-3: Yes, we have done the experiments at least three times. We have appended this information in figure legend.
Comment 2-4 There are no statistics for any of the blots shown.
Response 2-4: Thank you for your suggestion. We quantified the immunoblot band density and performed statistical analysis where necessary. Figures have been revised including statistical analysis information.
Comment 3: The manuscript is very poorly written with many grammatical errors, long vague, incorrect statements, and poor sentence construction throughout the text - too many to list. There are sloppy spelling mistakes which a simple spell check should pick up.
One example: starting line 29: According to a statistics, the incidence of about 1,735,350 new cancer cases and 609,640 cancer-related deaths have been recorded only in the United States, this year [2]. Does this mean that cancer is ‘only’ in the US??
Response 3: We have made extensive revision of the Abstract, Introduction and Discussion sections and paid attention to make grammatical and typographical errors in all through the manuscript.
Reviewer 2 Report
Manuscripts presents new knowledge. Although the experiments are sound but minor/major changes are required before publication:
1. Abstract contains materials and methods information. Please remove that and reframe the whole abstract.
2. Introduction section is small and does not present a precise earlier work done in current investigation. Authors need to provide some more background information.
3. Please use some other scheme to show significant differences in figure 2 and 3.
4. What control has been used in figure 2b during immunoprecipitation techniques?
5. Significant differences are needed in figure 4 for 0 and 6 hours of control in figure 4b.
6. Discussion is written like material and methods. Authors failed to discuss their results considering current available knowledge. Authors need to and a complete new thorough section of discussion.
Author Response
Point-by-Point Response to Reviewer’s Comments:
Manuscript ID ijms-405551
Title : p21WAF1/Cip1 regulation by hYSK1 activates SP-1 transcription factor and increases MMP-2 expression under hypoxic conditions
Reply to Reviewer #2
Manuscript presents new knowledge. Although the experiments are sound but minor/major changes are required before publication:
Comment 1: Abstract contains materials and methods information. Please remove that and reframe the whole abstract.
Response 1: Thank you for your kind suggestion. We have made extensive revision of the Abstract, eliminated all text related to methodology and made generalized statements highlighting the objective of the study and major outcomes.
Comment 2: Introduction section is small and does not present a precise earlier work done in current investigation. Authors need to provide some more background information.
Response 2: We have extensively revised the Introduction section, included background research and attempted to improve the coherence in the text flow.
Comment 3: Please use some other scheme to show significant differences in figure 2 and 3.
Response 3: Thank you for suggestion. We have changed the significant as shown ‘*’ in the statistical difference in Figure 2 and 3A.
Comment 4: What control has been used in figure 2b during immunoprecipitation techniques?
Response 4: As control of the proteins expression we have checked the expression of Myc (p21), V5 (hYSK1) and HA (SP-1) by western blot. As immunoprecipitation control we used normal IgG against hYSK1 or SP-1.
Comment 5: Significant differences are needed in figure 4 for 0 and 6 hours of control in figure 4b.
Response 5: Thank you for your suggestion. We have added the statistical significance between 0 and 6 hours in Figure 4b
Comment 6: Discussion is written like material and methods. Authors failed to discuss their results considering current available knowledge. Authors need to and a complete new thorough section of discussion.
Response 6: We have revised the Discussion section to its full scale and believe it is now acceptable.
Round 2
Reviewer 1 Report
Comments:
The positive – the results are very clear and nicely presented.
Negative – 1) There are many very careless errors throughout the manuscript. 2) The conclusions (or model) needs to be supported by at least one functional assay.
The manuscript needs very careful editing. There are too many grammatical errors, punctuation errors, throughout the manuscript, even after the “English editing”. Inappropriate use of past and present tenses need to be addressed. Some sentences are not comprehensible and need rewriting. Duplication of sentences and words. These are examples but not comprehensive:
Line 61: p21 WAF1/CIP1WAF1/CIP1
Lines starting 217: Not only does the sentence not make any sense it is repeated in line 229.
Line 32: Colloquialism: flashes the hopes for conquering cancer, needs to be changed.
Line 232: extra comma remove.
Line 232: We have previously reported that the other hands, ???????
The conclusions need more evidence-base. What do the cell look like under the microscope? Have the authors done any viability assays, migration assays (even basic scratch assays), proliferation assays, or any functional assays to support their model.
Author Response
Manuscript ID ijms-405551R1
Title : p21WAF1/Cip1 regulation by hYSK1 activates SP-1 transcription factor and increases MMP-2 expression under hypoxic conditions
Reviewer #1
The positive – the results are very clear and nicely presented.
Response: Thank you very much.
Negative –
Comment 1: There are many very careless errors throughout the manuscript.
Response 1: We have made further editing of the manuscript with great care and believe now the quality is improved.
Comment 2: The conclusions (or model) needs to be supported by at least one functional assay.
Response 2: Thank you for this suggestion. We have included results of cell migration asays in Fig. 5C and D to support the hypothesis. After knockdown of hYSK1, the cell migration was obviously decreased. Furthermore, when we silenced p21 WAF1/Cip1 in SK-MEL-28 melanoma cells, the cell migration was significantly increased under hypoxia conditions compared to normoxia and it was abolished by knockdown of hYSK1.
Comment 3: The manuscript needs very careful editing. There are too many grammatical errors, punctuation errors, throughout the manuscript, even after the “English editing”. Inappropriate use of past and present tenses need to be addressed. Some sentences are not comprehensible and need rewriting. Duplication of sentences and words. These are examples but not comprehensive:
Line 61: p21 WAF1/CIP1WAF1/CIP1
We have corrected this error
Lines starting 217: Not only does the sentence not make any sense it is repeated in line 229.
This sentence has been rewritten to improve clarity.
Line 32: Colloquialism: flashes the hopes for conquering cancer, needs to be changed.
We have done modifications of the word.
Line 232: extra comma remove.
We have corrected this error.
Line 232: We have previously reported that the other hands, ???????
Response 3: Thank you for careful reading and noticing this error. We have made appropriate corrections.
Comment 4: The conclusions need more evidence-base. What do the cell look like under the microscope? Have the authors done any viability assays, migration assays (even basic scratch assays), proliferation assays, or any functional assays to support their model.
Response 4: Thank you for your suggestion. The cell migration assay (Fig. 5C and D) showed that after knockdown of hYSK1, the cell migration was clearly decreased. Moreover, silencing of p21 WAF1/Cip1 in SK-MEL-28 melanoma cells resulted in significant increase in cell migration under hypoxia as compared to normoxia, and it was abrogted by knockdown of hYSK1.
Round 3
Reviewer 1 Report
Much improved.
Suggest line 33: One of the major reasons behind failure of anticancer therapies is the heterogeneous nature of the disease. Heterogeneity involves .....
Some minor formatting problems - line 83 spacing.
Line 191: We attempted. Your results show you did the IPs you didn't attempt them. Change to 'we assessed the interaction....
line 314: Conclusion is overstated. There are many mechanisms to an endpoint of cell function. Suggest the sentence is tempered: Taken together, we identified a potential regulatory mechanism, alternatively, we identified one of the regulatory mechanisms.
Author Response
Cover letter
December 26, 2018
Dear Managing Editor of International Journal of Molecular Sciences:
With this letter, we resubmit a manuscript entitled “p21WAF1/Cip1 regulation by hYSK1 activates SP-1 transcription factor and increases MMP-2 expression under hypoxic conditions” by Mee-Hyun Lee , Joydeb Kumar Kundu and Bu Young Choi to be considered for publication in International Journal of Molecular Sciences.
The hYSK1, a serine/threonine kinase (STK)-25, has been implicated in a variety of cellular functions including cell migration and polarity. We have recently reported that hYSK1 down-regulated the expression and functions of p16INK4a, a cell cycle regulatory protein, thereby enhancing migration and growth of cancer cells under hypoxic conditions. In this study, we further investigated the mechanisms underlying downregulation of p16INK4a and anti-migratory function of hYSK1. Our study revealed that p21WAF1/Cip1 is a novel binding partner of hYSK1. Moreover, the interaction between hYSK1 and p21WAF1/Cip1 led to the inhibition of SP-1 transcriptional activity, as revealed by a significant down-regulation of SP-1-mediated transactivation of p16INK4a promoter, and accelerated MMP-2 expression. Conversely, the knock-down of hYSK1 enhanced the p16INK4a promoter activity and protein expression, and diminished MMP-2 transcription and protein levels in hypoxic conditions as compared to control. Taken together, hYSK1 blocks the p21WAF1/Cip1 functions by direct interaction and inhibits the p16INK4a expression and induces MMP-2 expression by its regulations of SP-1 transcriptional activity under the hypoxia conditions.
We hope that you will consider this manuscript for publication in International Journal of Molecular Sciences.
Sincerely,
Bu Young Choi, Ph.D.
Professor
Pharmaceutical Science and engineering,
Seowon University, Cheongju, Chungbuk 361-742, South Korea.
Fax) +82 43 299-8410; Phone) +82 43 299-8411; E-mail) bychoi@seowon.ac.kr